# Insufficient: A scoping review of structural racism and intimate partner violence in US-based immigrant communities of color

Samantha Kanselaar[1]*, Sameera S. Nayak[2], Rochelle R. Davidson Mhonde[1], Areej Khan[1], Kyle Machicado[1], Jhumka Gupta[1]

1 Department of Global and Community Health, College of Public Health, George Mason University, Fairfax, Virginia, United States of America, 2 Department of Sociology, Anthropology, and Public Health, Center for Health, Equity, and Aging, University of Maryland Baltimore County, Baltimore, Maryland, United States of America

* skansela@gmu.edu

## Abstract

While there has been growing attention to research on structural racism and health inequities and the importance of naming structural racism as a driver of population health inequities, research focusing on structural racism and intimate partner violence (IPV) in United States (US)-based immigrant communities of color is lacking. This scoping review examined existing literature that names, operationalizes, and attributes structural racism as a determinant of IPV, and/or its health and social consequences among immigrant communities of color in the US. A search of eight databases (e.g., Medline, CINHAL) to identify studies related to IPV, structural racism, and immigrant communities retrieved 1896 articles. After independent reviewers screened papers for relevance, 32 articles were included in the scoping review. Articles were categorized into two categories for inclusion: Tier 1: Explicitly names structural racism as a driving factor of IPV and directly attributes IPV experiences or consequences to structural racism (articles that met all of our initial eligibility criteria); and Tier 2: Does not explicitly name structural racism as a driving factor of IPV experiences or consequences, but attributes IPV experiences or consequences to structural factors of oppression that align with themes of structural racism. Only one paper was identified that named and attributed IPV experiences to structural racism among US-based immigrants of color. Major research gaps in operationalizing, examining, attributing, and naming structural racism as a driver of IPV and sequelae in US-based immigrants of color persist. Findings serve as a call to action for scholars to move beyond solely traditional individual, cultural, and gender approaches and explicitly name and integrate structural racism into IPV research efforts among immigrant communities of color.

**Data availability statement:** All relevant data are within the paper and its Supporting Information files.

**Funding:** This work was supported by a George Mason University's MARIE program grant (awarded to SK under mentorship of JG). The funders had no role in study design, data collection and analysis, decision to publish, or preparation of the manuscript.

**Competing interests:** The authors have declared that no competing interests exist.

## Introduction

Immigrant communities comprise nearly 14% of the United States (US) population and contribute immensely to the vitality and development of the US [1], yet experience greater health and social inequities compared to US-born counterparts [2–10]. Although many immigrants typically arrive in the US with better health profiles compared to US-born individuals [11–13], this 'immigrant health advantage' erodes with greater time in the US [14,15]. Much of this health erosion may be due to increased exposure to structural stressors and the cumulative impact of health, social, and economic inequities (e.g., anti-immigrant policies, labor exploitation, financial disinvestment) [3–7,10,16].

For racialized immigrant communities, inequities are shaped at the intersection of race, ethnicity, assigned immigration status, and gender, producing unique vulnerabilities that differ from those faced by US-born racialized populations [3,6,17–19]. While US-born racialized populations also experience systemic barriers leading to inequities, US-based immigrants who are racialized as people of color (e.g., Latina/o, Asian, Arab, African, Caribbean) are uniquely impacted by such inequities [2,6,8–10]. This may be due to additional layers of mutually reinforcing systems of oppression and power through immigration-specific layers. For example, immigrants of color in the US are more likely to experience limited access to quality education, healthcare, housing, unequal employment opportunities, and higher levels of chronic diseases (e.g., hypertension, diabetes) and mental health challenges (e.g., anxiety, depression) compared to their US-born counterparts [2,6,7,10,12,16,20–22].

Structural racism encompasses relational systems of power that continuously reproduce and maintain racial hierarchies, privileging whiteness through the unequal distribution of resources, rights, and recognition [23–28]. It operates across mutually reinforcing systems (e.g., housing, education, employment, earnings, benefits, credit, media, healthcare, criminal justice, epistemic/knowledge production) as a key factor driving health and social inequities experienced by immigrants of color [3,6,23,24]. Drawing on Bailey et al. [23], we use structural racism as a broader term encompassing systemic and institutional forms of racism operating across mutually reinforcing systems.

Structural racism is enacted and reinforced through immigration and citizenship systems that formalize racial hierarchies and shape immigrants' access to rights, resources, and belonging. For example, upon arrival to the US, immigrants undergo formal racialization through immigration policies that assign racial identities (e.g., Sub-Saharan African immigrants often become racialized as Black, Middle Eastern and North African immigrants historically racialized as White) and legal status (e.g., temporary protected status, refugee, asylee, those living in the US without legal authorization, permanent residents) [6]. Additionally, these inequities may be compounded by various assigned immigration statuses. For example, those without legal authorization are more vulnerable to economic exploitation and reduced access to medical care [3,5,6]. Another enactment of structural racism occurs through informal racialization processes, such as immigration enforcement and criminalization (e.g., Immigration and Customs Enforcement raids), economic exploitation (e.g.,

wage theft), and systemic disinvestment (e.g., the legacy of redlining), further exacerbate these inequities [6]. The inequitable application of deportation is an enactment of structural racism that reflects this pattern, as U.S. immigration policy and enforcement has historically and contemporarily been used to exclude specific groups from residence, citizenship, and associated benefits by restricting citizenship to those classified as "white," constructing racial categories for specific groups, and allowing racial profiling to be sufficient grounds for immigration stops [6,28,29]. Consequently, these systemic processes can reinforce racial hierarchies in citizenship (e.g., anti-immigrant friendly policies), belonging (e.g., stereotypes, xenophobia), and health.

One pattern of health inequity that is of critical concern is the disproportionate, negative health and social impacts of male-perpetrated intimate partner violence (IPV) against women among US-based immigrant communities of color. IPV is any form of physical, sexual, emotional, or economic abuse by an intimate partner [30]. One in three women in the US has experienced at least one form of male-perpetrated IPV in their lifetime [31]. IPV consequences have been extensively documented and include poor mental health, negative maternal and child health outcomes, increased risk of sexually transmitted infections (including HIV/AIDS), injury, and death [32–35]. Findings from a recent systematic review showed that US-based immigrant women experience high levels of IPV, with past-year prevalence rates from community and/or clinic-based samples ranging from 4-47% and lifetime prevalence rates ranging from 14-93% [36]. Importantly, US-based immigrant women are disproportionately impacted by severe IPV consequences, including homicides, severe injury, post-traumatic stress disorder, and major depressive disorder [33,37–39]. The intersecting experiences of immigration status, race, ethnicity, and gender create a distinct context for IPV in immigrant communities of color. Immigrant women of color, for example, may navigate the combined impacts of gender-based discrimination, racialized systemic inequities, and immigration-specific vulnerabilities [3,6,17–19], such as restricted access to resources due to fear of deportation or language barriers [40,41].These intersecting experiences may contribute to heightened risks of IPV and more severe consequences, including increased social isolation, economic dependency, and barriers to seeking help [40,41].

Drawing on the scholarship of Black and transnational feminists, this review is guided by intersectionality as a core analytic framework, which conceptualizes power as relational and guides an examination of how multiple interlocking systems of oppression and power, such as gender inequity, immigration systems, and structural racism, shape lived experiences [17–19,23,42–46]. Through this framework, structural racism operates as a cross cutting system that interacts with gendered and immigration related hierarchies to reproduce and maintain compounded inequities [17–19,23,42–44]. A decolonial lens further extends this analysis by interrogating how Western-centric epistemologies and colonial logics shape what forms of knowledge are legitimized, whose experiences are centered, and whose are rendered invisible in IPV research [17–19,23,42–44]. Prior scholarship suggests that when structural racism is not explicitly named and attributed as a driver of inequities, such omissions reflect epistemic injustice (i.e., inequities in how knowledge is produced, justified, and validated) [17–19,23,42–44]. For example, Western-centered research paradigms rooted in colonial, gendered, and racial hierarchies privilege frameworks and data derived primarily from white women's experiences, while overlooking or inadequately capturing the realities of women of color, particularly immigrant women of color [17–19,23,42–44]. Recognizing this epistemic injustice underscores the need for this scoping review to understand how IPV scholarship has engaged with structural racism in IPV literature among US-based immigrants of color. This is an essential step toward guiding and transforming the knowledge base to better reflect the realities of US based immigrant women of color experiencing IPV.

Despite the disproportionate impacts of IPV in immigrant communities of color and increased recognition of naming and studying structural racism as it pertains to health and social inequities among immigrants of color [3,4,6,9,23,24,47,48], there has been little work explicitly examining the extent to which structural racism may have a role in perpetuating IPV experiences in US-based immigrant communities of color. For instance, structural racism in the form of anti-immigration policies can restrict access to employment and legal protections, perpetuate fear of deportation, foster financial dependency and entrapment, and can create environments conducive to IPV [40,41]. Economic exploitation is prevalent in industries relying on immigrant labor, with workers, particularly undocumented immigrants, experiencing wage theft and

unsafe conditions [23,24]. Simultaneously, disinvestment in infrastructure and social services in immigrant communities further perpetuates cycles of poverty and marginalization. These facets of structural racism may exacerbate IPV in immigrant communities by creating economic dependencies on abusive partners, social isolation, and restricted access to affordable housing and healthcare [41].

To further advance the IPV field, it is imperative to use an intersectional lens to systematically assess the current state of research on how structural racism impacts IPV experiences and outcomes within US-based immigrant communities of color [17–19,27,42]. By doing so, we can identify gaps in IPV literature where structural racism has not been explicitly defined, measured, examined, named, and attributed as a determinant of IPV. The findings from this scoping review will guide future research efforts, ensuring that such efforts are further equipped to address the root causes of IPV, leading to more effective and equitable interventions and policies tailored to immigrants of color. Such work is particularly important as immigrant communities are a priority population in the first-ever US National Action Plan to End Gender-Based Violence (GBV), necessitating critical research on how to address challenges faced by immigrant survivors [49]. Therefore, the goals of this scoping review are to 1) map the extent to which structural racism is examined, named, and attributed as an upstream factor impacting IPV experiences and consequences in US-based immigrant communities of color; 2) explore the depth of discussion regarding structural racism when the term was used; and 3) provide a roadmap for future work on operationalizing, naming, and attributing structural racism in IPV research focused on US-based immigrants of color. Failure to name structural racism as a determinant of health and social inequities perpetuates epistemic injustice and the marginalization of immigrant women's experiences. This work is critical to shifting the field from viewing IPV solely as a product of individual or cultural behavior to recognizing it as one rooted in systemic inequities. Explicitly naming structural racism enhances conceptual clarity, supports structural-level measurement and intervention, and ensures that research efforts address root causes rather than symptoms.

## Methods

### Design

This scoping review was guided by Arksey and O'Malley's framework and the preferred reporting items for systematic reviews and meta-analyses extension for scoping reviews (PRISMA-ScR; S1 PRISMA Checklist) [50,51]. A scoping review methodology was employed, as opposed to a traditional literature review, because of the conceptual complexity, interdisciplinary nature, and early stage of the literature in this area. Scoping reviews are particularly appropriate for mapping broad, heterogeneous bodies of literature and identifying conceptual gaps, especially when definitions, frameworks, and methodologies vary widely [50–52]. A traditional literature review would not have allowed us to capture this conceptual variability or to systematically assess how, and to what extent, structural racism is integrated into IPV research across disciplines.

### Search strategy

The research team developed our search strategy in collaboration with a community health subject librarian. We searched Medline, CINAHL, PyschInfo, ProQuest, Web of Science, SocIndex, Violence and Abuse Abstracts, and Social Science Database for relevant articles related to IPV and structural racism in immigrant communities of color from inception. Reference lists were also reviewed for additional relevant sources (i.e., backward citation chaining). This search was run in April 2025 using concepts broadly characterized as follows (Table 1): Concept 1: IPV. Any form of male-perpetrated IPV against women and girls, including physical, sexual, emotional, or economic was considered for this review. Umbrella terms such as "domestic violence", "gender-based violence", and "violence against women" were utilized to capture a wider range of literature on IPV; Concept 2: Structural racism. A comprehensive approach was utilized to capture themes of structural racism that align with our conceptualization of structural racism [3,6,23,24]. Although structural racism, systemic racism, and institutional racism may differ depending on definition used, they overlap and are often interrelated. For the purpose

**Table 1. Search concepts and keywords used.**

| Concept 1: Male-perpetrated IPV against women and girls (and) | Concept 2: Structural racism (and) | Concept 3: Immigrant (and) |
|---|---|---|
| *Keyword terms:*<br>• "Intimate partner violence" or<br>• "Domestic violence" or<br>• "Partner violence" or<br>• "Marital violence" or<br>• "Spousal abuse" or<br>• "Conjugal violence" or<br>• "Battered women" or<br>• "Gender-based violence" or<br>• "Violence against women" | *Keyword terms:*<br>• "Racism" or<br>• "Race" or<br>• "Discrimination" or<br>• "Redlining" or<br>• "Residential segregation" or<br>• "Systemic racism" or<br>• "Institutional racism" or<br>• "Immigration policies" or<br>• "Citizenship status" or<br>• "Immigration enforcement" or<br>• "Police brutality" or<br>• "Police violence" or<br>• "Incarceration" or<br>• "Deportation" or<br>• "Family separation" or<br>• "Neighborhood conditions" or<br>• "Economic exploitation" or<br>• "Eviction" or<br>• "Re-housing" or<br>• "Housing practices" or<br>• "Health care quality" or<br>• "Health care access" or<br>• "Racial profiling"<br>• "Xenophobia" or<br>• "Raids" | *Keyword terms:*<br>• "Immigrant" or<br>• "Migrant" or<br>• "Refugee" or<br>• "Asylee" or<br>• "Asylum seeker" or<br>• "Undocumented" or<br>• "Unauthorized" or<br>• "Foreign born" or<br>• "Alien" or<br>• "Illegal" |

of this review, the term structural racism will be used to capture all three forms guided by our definition and conceptualization of structural racism [3,6,23,24]; Concept 3: Immigrant communities of color. For the purpose of this review, the term "immigrant communities of color" encompasses all non-white individuals born outside of the US, regardless of immigration status (see Table 1 for more details). Immigrants who may be classified as 'white' despite racialization as 'non-white', such as Arab immigrants from the Middle East and North Africa, are included.

## Eligibility criteria

### Title and abstract screening

All titles and abstracts were reviewed independently for eligibility by two authors (SK and AK). To be considered for full-text review, the articles must 1) be peer-reviewed primary research studies; 2) focus on male- perpetrated IPV against women and girls; 3) include US-based immigrant communities of color; and 4) discuss themes of structural racism that align with our definition of structural racism indicated by the search strategy in Table 1. Studies progressed to full-text screening upon reviewer agreement, or after discrepancies were resolved by the full research team. Studies that neither clearly met nor failed to meet the inclusion criteria were sent to full-text review. This broad approach was taken to capture articles that may not explicitly name and attribute structural racism in the abstract but may do so in the full text. All systematic, scoping, or literature reviews were excluded, as were any editorials, commentaries, conference abstracts, protocols, books or book reviews.

## Full-text review

Full texts were reviewed initially by the same two authors (SK and AK). Inclusion and exclusion criteria used for the abstract and title screening continued to apply. In addition, articles were excluded if they did not explicitly name and attribute IPV, IPV-related help-seeking, or IPV-related health sequelae to structural racism. Upon completion of the full-text screening, only one article met our initial inclusion criteria. Thus, we broadened our inclusion criteria into two Tiers which had different objectives in how articles named, attributed, and operationalized structural racism as a determinant of IPV experiences or consequences and/or attributed structural racism to IPV experiences or consequences: Tier 1: Explicitly names structural racism as a driving factor of IPV and directly attributes IPV experiences or consequences to structural racism (articles that met all of our initial eligibility criteria); and Tier 2: Does not explicitly name structural racism as a driving factor of IPV experiences or consequences, but attributes IPV experiences or consequences to structural factors of oppression that align with themes of structural racism and related systems of oppression (e.g., structural violence, legal violence, structural barriers, systemic barriers, fear of deportation, etc.). Tier 2 included implicit references and attributions of structural racism. Articles that used the term 'systemic racism' that aligned with our conceptualization and definition of structural racism were included in Tier 1, while those that used the term with a different definition did not. This distinction was made to account for variations in how structural and systemic racism are defined and operationalized across the literature. Because the abstract review phase had already captured key study details related to the scoping review goals (e.g., population, IPV form, structural determinants), it was not repeated. Instead, the reviewers returned to the full texts to ensure that all eligible studies were reclassified appropriately under the expanded criteria. These two tiers yielded 32 articles (Table 2) [40,41,53–82]. To maintain rigor, two authors independently reviewed and categorized the full-text articles into the tiers (SK and KM). Through this process, eight articles were categorized differently between the two reviewers and were flagged for further discussion. There were two discrepancies between Tier 1 vs Tier 2 categorization, both ultimately categorized as Tier 2. There were also six discrepancies between Tier 2 categorization vs exclusion with one article being excluded and the remaining five articles included as Tier 2. The two reviewers met frequently to discuss decisions, and any discrepancies were resolved collaboratively through iterative dialogue, with the final decisions made by the senior author (JG) to ensure consistency and accuracy [83–86]. Through this process, we aimed to ensure conceptual consistency and reflexivity in identifying how structural racism was named, attributed, and operationalized across studies [83–86]. This collaborative and reflexive process aligns with best practices for critical scholarship guided by intersectional frameworks, where emphasis is placed on transparency and shared interpretation rather than statistical agreement [83–86].

## Data extraction and synthesis

Descriptive data were extracted and synthesized by one reviewer using narrative synthesis (Table 3). Narrative synthesis is an approach of synthesizing findings from multiple studies that relies on textual analysis of a study and is well-suited for analysis that requires a nuanced analysis of how complex factors such as structural racism are operationalized in the study [87]. Spot checks of data extraction and synthesis for 15 articles were performed by the full research team for quality assurance purposes. As our goals for the scoping review were not to synthesize study findings, descriptive study details related to our goals of mapping how structural racism is examined, named, and attributed in IPV literature, were extracted from included studies as reported (e.g., study design, immigrant country of origin, immigrant status, aim of study, type of IPV, form of structural racism). Key study findings can be found in (S1 Table).

## Results

Fig 1 displays the PRISMA diagram of the search process. The search returned 1896 records. After removing duplicates (n = 932), 964 remained for abstract and title screening. One hundred and nine articles met the eligibility criteria for full-text

**Table 2. Summary of the study title and aims, objectives, or purpose of the articles included in the scoping review.**

| Authors | Aims[a] |
|---|---|
| **Tier 1: Explicitly names structural racism as a driving factor of IPV directly attributes IPV experiences or consequences to structural racism** | |
| (Bhuyan & Velagapudi, 2013) [57] | "We examine how intersecting and interlocking oppressions shape the delivery of services to immigrant women who are facing violence and discuss what strategies advocates use to support women's safety and self-determination in an intense and at times hostile anti-immigrant environment." |
| **Tier 2: Does not explicitly name structural racism as a driving factor of IPV experiences or consequences, but attributes IPV experiences or consequences to structural factors of oppression that align with themes of structural racism (implicit references & attributions)** | |
| (Abboud et al., 2025) [53] | "Our study aimed to describe the different forms of IPV and understand factors that contribute to IPV from the perspectives of Arab American young adults in Chicagoland area." |
| (Abrego & Lakhani, 2015) [54] | [53]"By examining the integration prospects of immigrants in "liminal" legal standings beyond undocumented status but short of permanent residency, we demonstrate that even when they are legally present, the implementation practices of a multilayered immigration policy regime may cause them harm." |
| (Alsinai et al., 2023) [55] | "The goal of this study was to describe the lived experiences of DV victim-survivors who reported immigration related circumstances when petitioning for a domestic violence protection order (DVPO) in King County, WA." |
| (Bevilacqua et al., 2023) [40] | "To understand unique contexts for discrimination, violence victimization, and barriers to care for evidence-based programming and policy, this study examined experiences of violence and discrimination among Latino/a/x immigrants through in-depth qualitative interviews (IDIs) and one focus group discussion (FGD) conducted among Latino/a/x immigrants living in the state of Maryland and the District of Columbia." |
| (Bhuyan, 2008) [56] | "This article examines how the difference in signification (of terms) has direct social and political consequences with regard to who may access the benefits and protection offered to victims of domestic violence in the United States." |
| (Bryan et al., 2025) [58] | Descriptively analyze experiences of immigrant survivors of violence with police and how those experiences affect trust in police, government, and willingness to engage with institutions. |
| (Bui, 2003) [59] | "The present study examines help-seeking behavior among abused Vietnamese American women to understand factors associated with their decisions to seek help." |
| (Chenane & Pryce, 2024) [60] | "Thus, this study examines the impact of procedural justice and other cognate concepts on the obligation to obey and willingness to cooperate with the police in a sample of African immigrant women domiciled in the United States." |
| (Garni & Melander, 2023) [61] | "The purpose of this project is thus to explore how gender-based violence affects women's defensive asylum seeking." |
| (Gezinski & Gonzalez-Pons, 2021) [62] | "The purpose of this research study was to assess the challenges to service access and service delivery for survivors in Utah." |
| (Giordano et al., 2021) [63] | "This study examined Latino immigrant access to SAVAME-related services in Philadelphia, as perceived by providers working at Latino-serving organizations." |
| (Gray et al., 2024) [64] | "The current study aims to extend our current understanding of IPH by estimating the impact of firearm restrictions at the county level by employing proxies of state legislation." |
| (Ingram et al., 2010) [65] | "The purpose of this participatory action study was to document the experiences of Mexican immigrant women who filed VAWA self-petitions in two communities on the U.S.-Mexico border." |
| (Kaufman, 2024) [66] | "The purpose of this study is not to do a full test of either GST [General Strain Theory] or TAAO [Theory of African American Offending] or to conduct a theoretical competition between the two theories. Rather, the purpose is to focus on these two types of injustices, highlighted by both theories, because of their salience in the lives of Black and Latino men and the potential of these injustices to lead to IPV." |
| (Kimberg et al., 2021) [67] | "We sought to assess undocumented Latino immigrants (UDLI), Latino legal residents/citizens (LLRC) and non-Latino legal residents/citizens (NLRC) beliefs about disclosure of DV victimization to healthcare providers and healthcare provider reporting of DV to law enforcement and immigration authorities." |
| (López-Zerón et al., 2025) [68] | "The current study sought to answer the following community-driven research questions: (1) What are the lived experiences of Latina immigrant survivors in their search for housing in the US?, (2) How do Latina immigrant survivors define safe and stable housing?, and (3) What recommendations do Latina immigrant survivors have for direct service providers, funders, and technical assistance providers to improve existing housing supportive services?" |

*(Continued)*

**Table 2.** (Continued)

| Authors | Aims[a] |
|---|---|
| (Marrs Fuchsel, 2024) [69] | "This study examines immigrant Latinas' (ILs') help-seeking behaviors, types of support systems, and access to intimate partner violence (IPV) services during a global health crisis (COVID-19) at a community-based agency in a Northeastern state." |
| (Messing et al., 2015) [70] | "This research examines the impact of fear of deportation and trust in the procedural fairness of the justice system on willingness to report violent crime victimization among a sample of Latinas (N = 1,049) in the United States." |
| (Muchow & Amuedo-Dorantes, 2020) [71] | "The objective of this study is to explore the impact of immigration enforcement awareness on domestic violence calls in predominantly Latino noncitizen reporting districts." |
| (Nayak et al., 2023) [72] | "This study leverages focus groups with DV service providers to offer perspectives on how immigrant survivors navigate a hostile political climate and identifies structural challenges to their service utilization and well-being." |
| (Parson & Heckert, 2014) [73] | "This article also highlights some of the ways that care can function as biopolitical control and illuminates how a nuanced understanding of this phenomenon could help institutional practices." |
| (Raj A et al., 2005) [82] | "To explore forms of immigration-related partner abuse and examine the association of such abuse and immigration status with physical and sexual intimate partner violence (IPV) among South Asian women residing in greater Boston." |
| (Reina & Lohman, 2015) [74] | "The present study seeks to extend our understanding of domestic violence in the Latino population by exploring the experiences of Latina immigrants, predominantly Mexican women in a metropolitan area of Iowa, who faced domestic violence, had previously contacted an anti-violence organization, and had used its services at some point in their lives. Specifically, we examine the challenges they faced when they sought help from formal institutions and explore how immigration and domestic violence policies provide support to or endanger immigrant battered women" |
| (Rodriguez et al., 2018) [75] | "In this paper, the authors describe the process and results of a youth-led, qualitative study examining the impact of antiimmigrant policy on Latino families affected by DV in Georgia." |
| (Sabri & Campbell, 2024) [76] | "This study aimed to understand differing consequences of partners' firearm possession on abused women and barriers women face in reporting threats to safety due to the partners' possession of a firearm. Additionally, the study explored participants' perceptions of effective approaches to risk assessments and safety planning with women who are at-risk for being harmed by their partners' possession of a firearm." |
| (Singh & Bullock, 2020) [77] | "This study examined race, class, gender, and their intersections in newspaper coverage of the 2013 reauthorization of the Violence Against Women Act (VAWA)." |
| (Solis & Heckert, 2021) [41] | "This article explores how experiences of intimate partner violence (IPV) combined with immigration-related stress shaped pregnancy and postnatal experiences among two immigrant women in the U.S.-Mexico border region of El Paso, TX." |
| (Valdovinos et al., 2021) [78] | "This article intends to answer the following research question: Research Question: How do Latina undocumented immigrant IPV survivors describe the role that their social identities play when they experienced barriers when seeking help for the IPV." |
| (Valdovinos & Vanegas, 2024) [79] | "Sixteen interviews with Mexican undocumented immigrant women who experienced intimate partner violence (IPV) were analyzed to investigate how the women perceived that their immigration status affected their seeking social services for their mixed-status families." |
| (Yuan et al., 2022) [80] | "Using a representative sample from a major US city, we address three key questions to better understand the relationship between race/ethnicity, immigration generational status, procedural justice, and willingness to report domestic violence to the police. First, we examine whether generational status is associated with domestic violence reporting. Specifically, we hypothesise that first-generation immigrants will be less likely than second-and-third or greater-generation immigrants to report domestic violence in their communities to the police. Second, we examine whether there will be variation between immigrant subgroups (e.g., Mexican, Chinese, and Vietnamese) in their willingness to report domestic violence to the police. Finally, we investigate whether procedural justice is associated with immigrants' willingness to report domestic violence." |
| (Zero et al., 2023) [81] | "This research aims to address a critical gap in research by reporting the unique barriers that prevent undocumented monolingual Spanish-speaking immigrant women from disclosing IPV." |

[a]Direct quotes are presented with quotation marks

**Table 3. Study characteristics of the articles included in the scoping review.**

| Study | Method | Study Design/framework | Data Type | Participant sample size (N) | Immigrant countries of origin as described by authors[a] | Immigrant category included as described by authors | Immigrant Generation | Form of IPV | Key structural factor(s) that align with structural racism as a driving |
|---|---|---|---|---|---|---|---|---|---|
| Tier 1: Explicitly names structural racism as a driving factor of IPV and directly attributes IPV experiences or consequences to structural racism as a driving factor. | | | | | | | | | |
| (Bhuyan & Velagapudi, 2013) [57] | Qualitative | Community-based research methods | • Focus group discussions<br>• Individual interviews | 24 | Not specified | Not specified | Not specified | Not specified (focuses on domestic violence and sexual assault services) | • Illegality<br>• Deportability<br>• Immigration policy |
| Tier 2: Does not explicitly name structural racism as a driving factor of IPV experiences or consequences, but attributes IPV experiences or consequences to structural factors of oppression that align with themes of structural racism (implicit reference and attributionsS) | | | | | | | | | |
| (Abboud et al., 2025) [53] | Qualitative | Community based participatory research | Individual interviews | 22 | Most born in the US but described family origins from<br>• Palestine<br>• Jordan<br>• Lebanon<br>• Syria<br>• Egypt<br>• Sudan<br>2 born outside of US:<br>• Jordan<br>• Palestine | Not specified | Self-identified as first- (born outside of the U.S.) or second generation (born in the U.S.) | • Sexual<br>• Physical<br>• Psycho-logical | • Xenophobia<br>• Arab racism |
| (Abrego & Lakhani, 2015) [54] | Qualitative | Legal Violence Framework | Individual interviews | 108 | • El Salvador<br>• Mexico<br>• Central American<br>• Guatemala | Immigrants who have been granted humanitarian relief:<br>• U Visa holders, beneficiaries of the Violence against Women Act provisions<br>• political asylees<br>• Temporary Protected Status recipients | Not specified | Domestic violence | Immigration policies |

*(Continued)*

| Study | Method | Study Design/ framework | Data Type | Participant sample size (N) | Immigrant countries of origin as described by authors[a] | Immigrant category included as described by authors | Immigrant Generation | Form of IPV | Key structural factor(s) that align with structural racism |
|---|---|---|---|---|---|---|---|---|---|
| (Alsinai et al., 2023) [55] | Qualitative | Textual analysis | Domestic violence protection order narratives | 39 cases | Information on petitioners was not available. | Information on petitioners was not available. | Information on petitioners was not available. | • Physica<br>• Sexual<br>• Psychological coercion | • Interconnected social structures and systems of oppression<br>• Fear of deportation through individual threat<br>• Workplace authorization<br>• Family separation |
| (Bevilacqua et al., 2023) [40] | Qualitative | Not specified | • Focus group discussions<br>• Individual interviews | 22 | • Colombia<br>• Dominican Republic<br>• Ecuador<br>• El Salvador<br>• Honduras<br>• Mexico<br>• Nicaragua | • Asylee<br>• Refugee<br>• Permanent resident<br>• Undocumented<br>• Visa holder | Not specified | • Physical and/or sexual Intimate partner violence<br>• Work place violence | • Fear of deportation<br>• Family separation<br>• Fear of retribution |
| (Bhuyan, 2008) [56] | Qualitative | Ethnographic | • Discourse analysis of immigration policy and immigrant provisions in federal violence against women legislation<br>• Participant observation at Chay<br>• Individual interviews<br>• Community organizing and advocacy texts generated by domestic violence and immigrant rights' organizations circulated online through Web pages, e-mailed as action alerts and policy briefs, and distributed during conferences and trainings on advocacy with immigrant survivors | 14 | South Asian | Not specified | Not specified | • Domestic violence<br>• Partner/ spousal abuse | Immigration policies |

*(Continued)*

| Study | Method | Study Design/framework | Data Type | Participant sample size (N) | Immigrant countries of origin as described by authors[a] | Immigrant category included as described by authors | Immigrant Generation | Form of IPV | Key structural factor(s) that align with structural racism |
|---|---|---|---|---|---|---|---|---|---|
| (Bryan et al., 2025) [58] | Qualitative | Community based participatory research | Individual interviews | 17 | • Brazil • Bangladesh • The Dominican Republic • Guatemala • Honduras • India • Mexico • Pakistan • Thailand • The United Arab Emirates • Venezuela | • Legal status • No legal status • Pending applications | Not Specified | • Financial • Emotional • Psychologic • Spiritual • Physical • Sexual | • Fear of deportation • Racialized criminalization |
| (Bui, 2003) [59] | Qualitative | Feminist Tradition | Individual Interview | 34 | Vietnam | Not specified | Not specified | Not specified | • Immigration law • Police brutality • Racial profiling |
| (Chenane & Pryce, 2024) [60] | Quantitative | Cross-sectional | Secondary survey data | 478 | Kenya | • US citizen • Not US citizen, permanent resident • Not permanent resident | Not specified | Not specified | • Police brutality • Procedural justice/injustice • Distributive justice/injustice |
| (Garni & Melander, 2023) [61] | Qualitative | Ethnographic | • Ethnographic observations • Research reports written by humanitarian organizations conducting on-the-ground work along the United States-Mexico border. • Oral history interviews with attorneys who regularly work directly with asylum seekers in detained and non-detained settings in the United States | 11 | • Honduras • El Salvador • Guatemala • Mexico • Brazil • Additional countries not specified | Defensively seeking asylum | Not specified | Domestic violence | Asylee law enforcement |
| (Gezinski & Gonzalez-Pons, 2021) [62] | Qualitative | Grounded Theory | • Focus group discussions • Individual interviews | 102 | Not specified | Not specified | Not specified | Intimate partner violence | Fear of deportation |

(Continued)

**Table 3.** (Continued)

| Study | Method | Study Design/framework | Data Type | Participant sample size (N) | Immigrant countries of origin as described by authors[a] | Immigrant category included as described by authors | Immigrant Generation | Form of IPV | Key structural factor(s) that align with structural racism |
|---|---|---|---|---|---|---|---|---|---|
| (Giordano et al., 2021) [63] | Quantitative | Cross-sectional | Primary survey data | 43 | Latino | Not specified | Not specified | Not specified (focuses on DV services) | • Police brutality Anti-immigrant climate • Xenophobe rhetoric • Police brutalities • Fear of deportation |
| (Gray et al., 2024) [64] | Quantitative | Cross-sectional | Secondary survey data | 2,668 | • White (non-Latina) • Black (non-Latina) • Latina | Not specified (does include non-citizens and people with limited English speaking) | Not specified | • IPV legislation • Intimate partner homicide | • Firearm legislation IPV legislation • Economic marginalization |
| (Kaufman, 2024) [66] | Quantitative | Cross-sectional | Secondary survey data | 1,679 | • Black • Latino | Not Specified | Not Specified | IPV perpetration | Disparate treatment from governmental authorities such as the police |
| (Ingram et al., 2010) [65] | Qualitative | Participatory action research | Individual Interview | 21 | Mexican | Not specified | Not specified | Not specified | • Fear of deportation • Immigration law penalties |
| (Kimberg et al., 2021) [67] | Quantitative | Cross-sectional | Primary Survey data | 667 | Latino | Undocumented | Not specified | Not specified | Fear of deportation |
| (López-Zerón et al., 2025) [68] | Qualitative | Participatory Action Research | Individual Interviews | 14 | • Mexico • Guatemala • El Salvador • Bolivia | Not specified but does include examples of narratives from those without legal status | Not specified | Not specified | • Landlord exploitation (e.g., price gouging, deportation threats, unsafe living conditions) • Immigration policies |
| (Marrs Fuchsel, 2024) [69] | Qualitative | Not specified | Individual Interview | 19 | • Brazil • Columbia • Dominican Republic • Ecuador • El Salvado • Guatemala • Mexico • Guatemala • Dominican Republic | Not specified | Not specified | Not specified | Fear of deportation |

*(Continued)*

| Study | Method | Study Design/framework | Data Type | Participant sample size (N) | Immigrant countries of origin as described by authors[a] | Immigrant category included as described by authors | Immigrant Generation | Form of IPV | Key structural factor(s) that align with structural racism |
|---|---|---|---|---|---|---|---|---|---|
| (Messing et al., 2015) [70] | Quantitative | Cross-sectional | Secondary survey data | 1049 | Latinas | Not specified | Not specified | Not specified | • Police brutality Fear of deportation<br>• Procedural fairness<br>• Systemic biases |
| (Muchow & Amuedo-Dorantes, 2020) [71] | Quantitative | Cross-sectional | Secondary data | 53,520[b] | Not specified, but states predominately Latino non citizen neighborhood | Not specified | Not specified | Domestic violence | Immigration enforcement |
| (Nayak et al., 2023) [72] | Qualitative | Commission on Social Determinants of health Ecosocial Theory of Disease Distribution | Focus group discussions | 38 | Not applicable (DV service providers) | Not applicable (DV service providers) | Not applicable (DV service providers) | Not applicable (DV service providers) | • Anti-immigrant hostile political environment Structural barriers in the process of U-visa and VAWA petitions (social isolation, dehumanization) |
| (Parson & Heckert, 2014) [73] | Qualitative | Not specified | Individual Interviews | 98 | • Predominantly Mexico<br>• Other countries not specified | Unauthorized | First or 1.5 generation immigrants | Intimate partner violence | • Immigration policies |
| (Raj A et al., 2005) [82] | Quantitative | Cross-sectional | Primary survey data | 189 | • Indian<br>• Bangladeshi<br>• Pakistani<br>• Sri Lankan<br>• Nepalese | Not specified | Not specified | Physical and/ or sexual Intimate Partner violence | • Immigration policies Fear of deportation |
| (Reina & Lohman, 2015) [74] | Qualitative | Intersectionality and Feminist approach | • Focus group discussions<br>• Individual interviews | 10 | Latina immigrants (Mexican, Central/ South American) | • Undocumented at time of IPV experience<br>• 9 were in process of adjusting status through U-Visa | Not specified | Domestic violence | • Institutional discrimination<br>• Unstable residency<br>• Economic inequality<br>• Immigration law enforcement |
| (Rodriguez et al., 2018) [75] | Qualitative | Participatory action research | Individual Interviews | 18 | • Mexico<br>• Central America<br>• South America | Not specified | Not specified | Domestic violence | • Anti-immigrant sociopolitical climate<br>• Immigration policies |

*(Continued)*

| Study | Method | Study Design/framework | Data Type | Participant sample size (N) | Immigrant countries of origin as described by authors[a] | Immigrant category included as described by authors | Immigrant Generation | Form of IPV | Key structural factor(s) that align with structural racism |
|---|---|---|---|---|---|---|---|---|---|
| (Sabri & Campbell, 2024) [76] | Qualitative | Not specified | Individual Interviews | • 45 immigrant survivors of IPV • 17 service providers serving survivors of IPV | • Africa • Asia • Caribbean • Latin America | Not specified | Not specified | Not specified | • Fear of deportation |
| (Singh & Bullock, 2020) [77] | Qualitative | Feminist discourse | Newspapers | 93[c] | Not specified | Undocumented | Not specified | Not specified | • Criminal justice regarding the Violence against women act • Fear of deportation |
| (Solis & Heckert, 2021) [41] | Qualitative | Case study | Individual interview | 2 | • Mexico • South America • Africa | Not specified | Doesn't specify for the subgroup of 5 but larger group-included first and second generation | Not specified | Immigration law enforcement |
| (Valdovinos et al., 2021) [78] | Qualitative | Intersectional Chicana feminist approach testimonio methodology | Individual Interviews | 20 | • Mexico • El Salvador • Guatemala • Honduras • Nicaragua | Undocumented at time of IPV experience | Not specified | Not specified | • Fear of deportation • Racial profiling |
| (Valdovinos & Vanegas, 2024) [79] | Qualitative | Testimonio methodology | Individual Interviews | 16 | Mexico | Undocumented | Not specified | Not specified | • Immigration law • Fear of deportation |
| (Yuan et al., 2022) [80] | Quantitative | Cross-sectional | Secondary survey data | 3756 | Not specified | Not specified | First, second, and third or greater generations | Willingness to report domestic violence | • Procedural justice • Immigration status |

*(Continued)*

PLOS Global Public Health

Global Public Health

**Table 3.** (Continued)

| Study | Method | Study Design/framework | Data Type | Participant sample size (N) | Immigrant countries of origin as described by authors[a] | Immigrant category included as described by authors | Immigrant Generation | Form of IPV | Key structural factor(s) that align with structural racism |
|---|---|---|---|---|---|---|---|---|---|
| (Zero et al., 2023) [81] | Qualitative | Ethnography | • Longitudinal participant observation<br>• Study of legal affidavits written by undocumented Latina victims about their experiences of IPV<br>• Individual interviews | 14 | Spanish speaking | Undocumented | Not specified | Not specified | Fear of deportation |

[a]If authors included race/ethnicity, this is included when countries of origin were not specified.

[b]Observations: calls for service dispatched to Los Angeles Police Department patrols from 2014 through 2017.

[c]Articles: newspaper articles and editorials published in high-circulating newspapers between March 28, 2012 and March 7, 2013 were examined.

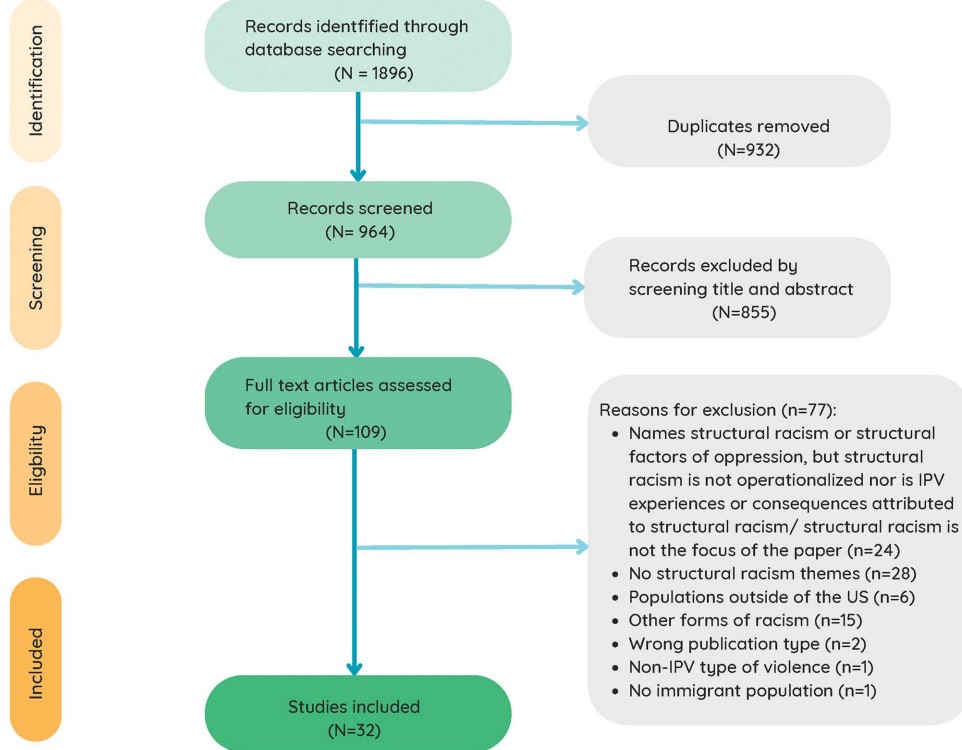

**Fig 1. PRISMA-ScR flowchart of literature search and data collection process.**

review. We excluded 77 articles for the following reasons: no structural racism themes (n = 28); population outside of the US (n = 6); other forms of racism (e.g., interpersonal) (n = 15); wrong publication type (n = 2); non-IPV type of violence (n = 1); no immigrant population (n = 1). A total of 32 articles are included in this review.

Of note is that we identified 24 additional articles that explicitly referenced the term structural racism or related structural factors in their introductions when justifying their research argument or in their discussion sections for future research directions (Fig 1) [88–111]. Given that these articles did not focus on operationalizing or attributing IPV or IPV consequences to structural racism or related structural factors, they were not included in the final articles for this scoping review.

Table 3 summarizes results by article characteristics. Across 32 studies, nearly all articles were qualitative (n = 23) [40,41,53–59,61,62,65,68,69,72–79,81], while the remaining were quantitative (n = 9) [60,63,64,66,67,70,71,80,82]. Most studies included Latina/o or Hispanic immigrant populations from predominately Latin, South, and Central America (n = 18) [40,41,54,58,61,63–65,67–70,73–76,78,79], with fewer including various Asian (n = 5) [56,58,59,76,82], Middle Eastern and Arab (n = 2) [53,59], Caribbean (n = 1) [76], and African countries of origin (n = 3) [41,60,76], while eight did not explicitly specify immigrant countries of origin or this information was not accessible [55,57,62,71,72,77,80,81]. The form of IPV included in studies was often not specified or measured directly (n = 17) [41,57,59,60,63,65,67–70,72,76–81], although some broadly included the terms domestic violence (n = 13) [54–58,61,67,71,74,75], or IPV (n = 2) [62,73], IPV perpetration (n = 1) [66], intimate partner homicide (n = 1) [64], IPV legislation (n = 1) [64], and willingness to report domestic violence (n = 1) [80]. Forms of IPV were specified in five articles [40,53,55,58,82]. The most common theme of structural racism mentioned was fear of deportation. Further study characteristics are detailed in Table 3.

**Tier 1: Directly attributes IPV experiences or consequences to structural racism and explicitly names structural racism as a driving factor.**

In all, only one article (3%) was identified that explicitly named structural racism (referred to by authors as systemic racism) and attributed IPV or IPV consequences to structural racism [57]. The study employed an intersectionality framework to explore how interlocking systems of oppression shape the delivery of services to immigrant women, specifically within a hostile anti-immigrant environment [57]. By addressing the systemic barriers and inequities faced by these women and the explicit role played by structural racism as a significant barrier for seeking support, the paper acknowledged the role of structural racism and also provided a comprehensive analysis of its impact on IPV and the strategies advocates use to support affected individuals [57].

**Tier 2: Does not explicitly name structural racism as a driving factor of IPV experiences or consequences, but attributes IPV experiences or consequences to structural factors of oppression that align with themes of structural racism (e.g., implicit references and attributions).**

Tier 2 included 31 articles (97%) [40,41,53–56, 58–82]. These articles did not name or attribute structural racism as a determinant of IPV or IPV consequences. However, they implicitly referenced or attributed IPV and IPV consequences to structural factors that align with themes of structural racism (e.g., fear of deportation, economic dependency, racial profiling, police brutality). For example, when discussing findings such as fear of deportation, articles commonly referred to implicit terms like legal violence, structural violence, or institutional, structural, or systemic barriers in place of explicitly naming structural racism when later analyzing, contextualizing, and/or discussing such findings. Articles in Tier 2 may also have interpreted and attributed findings such as fear of deportation within the framework of immigration laws and policies or anti-immigrant environments rather than stemming from structural racism.

## Discussion

To our knowledge, this is the first scoping review to examine the extent to which structural racism is named, examined, and discussed as a determinant of IPV experiences and consequences in US-based immigrant communities of color. Although our findings acknowledge some recognition of how structural racism factors may influence IPV and its harmful repercussions, significant gaps persist in adequately defining, measuring, attributing, and naming structural racism as a determinant of IPV. The scoping review findings underscore the necessity to integrate structural racism as a fundamental component in the research and analysis of IPV within US-based immigrant communities of color. Importantly, naming structural racism is not solely a semantic choice; it is an act of epistemic justice that works to make systems of power and oppression visible and, therefore, open to critique and accountability [17–19,23,42–44]. Without naming and attributing it, structural racism remains unmeasured, under-examined, and unchallenged, allowing its influence to persist unchecked across research, policy, and practice [17–19,23,42–44].

Despite calls from immigrant health and anti-racism scholars to shift public health discourse towards examining systems of oppression and power [2,4,6,7,9,23,24,28,29,46], the findings of our scoping review reinforce that the IPV and immigrant literature continues to predominantly center individual behavior and cultural explanations often without explicit consideration of overlapping systems of oppression and power, such as structural racism [17,19,42,43,45,46,112–116]. Gender inequity has also long been recognized as the most critical system of oppression and a central focus in IPV literature, leading to valuable insights and significant progress in the field. However, when considering the impacts of IPV among immigrant communities of color, solely examining and intervening on gender inequity without also addressing additional and overlapping systems of oppression and power, such as structural racism is not sufficient and limits the field's capacity to fully capture the complex realities shaping survivors' experiences [17,19,42,43,45,46,112–116]. Our tiered analysis offers additional evidence of this gap with only one study that explicitly named and attributed structural racism as a key driver of IPV (Tier 1), while many others (Tier 2) implicitly referenced structural conditions (e.g., legal violence, structural violence) but fell short of making this conceptualization or attribution explicit. It should be noted that some Tier

2 studies may reflect structural silencing rather than conceptual absence, wherein external, institutional, or disciplinary constraints prevent researchers from explicitly naming and attributing structural racism [117,118]. While this possibility cannot be fully assessed within the scope of this review, it warrants consideration in future work, particularly as publication bias and structural silencing may become increasingly salient in shaping how scholars frame structural racism as a determinant of IPV [117,118]. Our findings underscore the need for IPV researchers to move beyond partial or implicit engagement and to explicitly define, measure, attribute, and name the influence of structural racism, alongside gender inequity, individual, and cultural factors, to more accurately understand and address IPV among U.S.-based immigrant communities of color.

Misra and colleagues' framework offers a critical foundation for understanding how structural racism operates through policy, institutional practices, and social stratification to shape immigrant health inequities [6]. Many of these same inequities, such as economic marginalization, housing instability, and legal precarity, are also documented drivers of IPV [6,56,119–121]. However, our scoping review findings reveal that structural racism was rarely defined or examined as a primary analytic construct in IPV and immigrant literature (Table 3). In studies categorized as Tier 2 of our findings (Table 3), structural racism surfaced indirectly through emergent themes (e.g., fear of deportation, immigration enforcement, housing discrimination) rather than being named, theorized, or operationalized as an explicit exposure. To address these critical gaps identified by the findings of our scoping review, we encourage IPV researchers to extend the findings of Misra and colleagues and apply this framework along with intersectionality and decolonial frameworks to IPV research within US-based immigrant communities of color, shifting from individual behavior and cultural explanations to a more nuanced analysis of systemic inequities [17–19,23,42–44]. Importantly, gender inequity and structural racism are not separate forces but mutually reinforcing systems of oppression that must be examined together to fully capture the complexities of IPV among immigrants of color [17–19,23,42–44]. Integrating these frameworks can illuminate how structural forces such as immigration enforcement, racialized legal exclusions, labor exploitation, and housing precarity compound IPV risk and limit survivors' access to support. By advancing intersectional, structurally informed approaches, IPV research can better inform policies and interventions aimed at disrupting these upstream drivers and promoting justice for US-based immigrant survivors of color.

To advance this work, future IPV research must prioritize the explicit operationalization and examination of structural racism as an exposure within intersectional and decolonial frameworks [17–19,23,42–44]. Such approaches will help research move beyond approaches where it emerges thematically to approaches that intentionally define, measure, and analyze it as a central construct [3,6,27,28,122–124]. Our scoping review found that while structural racism often surfaced through themes such as fear of deportation, racialized immigration enforcement, and housing discrimination, these dynamics were rarely named or examined as manifestations of structural racism (Tier 2 findings in Table 3). Instead, they were frequently described as general "structural factors," "systemic discrimination," or "systemic violence" with no clear attribution to structural racism. This lack of explicit framing and examination reproduces and maintains epistemic injustice by upholding colonial, racialized, and gendered research frameworks that privilege Western-centric epistemologies and marginalize immigrant survivors of color [17–19,23,42–44]. This continued erasure of structural racism as a root cause of IPV in immigrant communities of color and limits the field's ability to develop interventions and policy that target its underlying drivers. Naming structural racism, rather than vaguely referencing structural conditions, is essential for advancing conceptual clarity, analytic precision, and policy accountability [3,6,27,28,122–124]. Without intentional definitions and measurements of structural racism in IPV research, we risk overlooking the deep-seated systemic barriers that perpetuate IPV and its consequences among immigrants of color (e.g., immigration enforcement and criminalization, economic exploitation, systemic disinvestment, anti-immigrant policies, xenophobia) [3,6,27,28,122–124].

Importantly, we must adopt a comprehensive approach to understanding structural racism for US-based immigrant communities of color, recognizing that it is reinforced by interconnected systems such as immigration systems, housing, education, employment, income, benefits, credit, media, healthcare, and criminal justice [3,6,23,28,45,46]. Our findings

underscore that much of the current literature on IPV and immigrant communities discusses findings such as the 'fear of deportation' as both a barrier to help-seeking and/or a mechanism that perpetuates IPV (Table 3). While continuing to advance research in this area, it is imperative to also consider additional forms of structural racism that immigrants of color experience as an exposure. Future work is needed to examine the impacts of discriminatory housing policies and neighborhood disadvantage, which shape the environments where immigrant communities live and can exacerbate stressors that contribute to IPV and its consequences [62,78,92]. Specifically, it is crucial to distinguish between 'immigrant enclaves' (i.e., a high number of ethnically similar people within a defined geographic area) and immigrants who are forced into disinvested neighborhoods through discriminatory housing practices, as these environments present different challenges. Although policies aiming to protect survivors from housing discrimination exist, they may not equitably serve immigrant communities of color [62,78,92]. Research shows that survivors of color face higher rates of housing discrimination compared to white survivors, and this may be further compounded by immigration status (e.g., refugee, unauthorized, permanent resident) [62,78,92,125]. Therefore, it is essential to examine the intersectional experiences of racialized groups, taking into account their varied immigration statuses, to better understand and address these disparities.

Additionally, our findings highlight a critical gap in how factors of structural racism such as labor market discrimination, limited access to healthcare (e.g., help-seeking), immigration enforcement and criminalization, and racial profiling within the criminal justice system may further compound vulnerabilities to IPV [40,59,67,70,72,79]. While these factors often emerged thematically in the literature, few studies explicitly examined them as manifestations of structural racism or analyzed how they operate at the intersection of gender and immigration status to compound IPV risk. Research has shown that these systemic inequities may further compound vulnerabilities to IPV by limiting economic opportunities, increasing stress, and restricting access to support services, but our review reveals that the underlying mechanisms remain insufficiently explored in immigrant communities of color [40,59,67,70,72,79]. Broadening the scope of research to include a comprehensive approach operationalizing structural racism can contribute to a more nuanced understanding of the mechanics underlying this relationship. This, in turn, can 1) help establish rigorous datasets that can link IPV with area-level variables for structural racism and 2) inform tailored and more effective interventions by integrating a structural racism lens into ongoing gender inequity interventions aimed at addressing IPV among US-based immigrants of color. Importantly, IPV and or immigration variables are lacking in many nationally representative datasets making it challenging to link area-level measures with IPV in immigrant communities of color. More efforts are needed for IPV variables to be ethically and safely collected as part of nationally representative datasets, as well as the inclusion of immigration variables (e.g., US vs non US born) [126]. To achieve this, dedicated funding streams are essential to support research that prioritizes the intersection of structural racism, gender inequity, and immigration (Table 4).

Recent research has recommended novel methodological approaches for measuring structural racism. Hardeman et al. [124] recommend measurements focused on historic, geographic, and ecological manifestations of structural racism with measures including historical prevalence of enslavement, lynching, and other manifestations of racialized violence, prevalence of discriminatory federal, state, and local policies, geographic segregation, measures of redlining, and the use of latent variable approaches to measure multifaceted exposures of structural racism [124]. Gee and Ford [28] have identified multiple dimensions of structural racism that should be examined as fundamental causes of health disparities, including social segregation, immigration policy, historical factors, and examples of structural racism in cyberspace [28]. Krieger [27] recommends measures that can be categorized as "explicit" (laws, policies, and rules explicitly designed to discriminate against target groups), "non-explicit" (laws, policies, and rules designed to evade antidiscrimination laws to discriminate against target groups), or area-based or institutional non-rule measures regarding legacies and indicators of structural injustice (population-based data quantifying the disparities between the targeted group the privileged group, the prevalence of discriminatory institutional or public practices, or the presence of public monuments or commemorations supporting racism) [27]. To facilitate this research, national data systems should begin collecting data on these measures, and funders should require proposals to outline how structural racism will be conceptualized and operationalized.

 

**Table 4. Recommendations for practice, policy, and research.**

| Practice |
| --- |
| • **Integration of structural racism lens:** Integration of navigating structural racism into gender equity interventions among immigrant communities of color is paramount. |

| Research |
| --- |
| • **Integration of structural racism as a fundamental component:** IPV researchers to integrate structural racism as a fundamental component in the analysis of IPV among immigrant communities of color and offers a roadmap for future work. |
| • **Operationalize and Measure Structural Racism:** Future research must focus on developing robust methodologies to operationalize and measure structural racism in relation to IPV among immigrant communities of color. This will provide clearer insight on how structural racism exacerbates IPV and inform tailored interventions. |

| Policy |
| --- |
| • **Ethical Inclusion of IPV in National Datasets:** There is a critical need for IPV variables to be ethically and safely collected as part of nationally representative datasets. |
| • **Inclusion of Immigration variables in National Datasets:** There is a critical need for immigration variables to be ethically and safely collected as part of nationally representative datasets. |
| • **Research funding:** Dedicated funding is urgently needed to support comprehensive research that examines structural racism and IPV among US-based immigrants of color. Funding should be directed towards the development of community-engaged research and interventions that are culturally responsive to the needs of immigrant survivors of color. This includes supporting partnerships with immigrant communities to ensure that research and interventions are relevant, impactful, and sustainable. |

Research that explicitly operationalizes and examines measures of IPV and structural racism among US based immigrants could guide the development of programs and policy, including the development of prevention frameworks that account for stressors related to structural racism, survivor services that provide legal and immigration support to help survivors navigate discriminatory systems, and policies that aim to support immigrant survivors by directly addressing barriers to seeking help posed by structural racism.

It should be noted that most of the articles reviewed in the scoping review were focused on Latino immigrant populations, reflecting a systematic underrepresentation of additional immigrant populations. While synthesizing differences in the experiences of structural racism and IPV among immigrant groups goes beyond the scope of this review, many migrant populations have been shown to experience similar structural stressors [127,128]. However, future research on the role of structural racism as a driver of IPV should examine differences across immigrant groups, form of IPV, and manifestation of structural racism. Moreover, research funding structures within the US generally have under-funded immigrant health research, and existing research tends to focus on one immigrant population at a time. These factors also contribute to the relatively larger body of research on Latino immigrant communities versus additional immigrant populations. As disparities are often driven not solely by ethnic origin, but also by an individual's position in the local social hierarchy, future research should examine differences within immigrant groups, as immigrants who are darker skinned or racialized as Black could face greater discrimination [28,129]. Future research is also needed to capture how structural racism manifests to impact IPV among immigrants holding various immigration statuses (e.g., refugee, asylee, without legal authorization, special immigrant visa), as our review noted significant gaps in immigrant categories examined. This variation is important because assigned immigration status can shape exposure to structural racism and IPV in distinct ways. For instance, many immigrant women in the US are forcibly displaced, and stressors associated with forced displacement (e.g., financial strain, loss, social isolation, gender role changes) can exacerbate vulnerability to IPV [57,119,130–132]. Additionally, some immigrants of color may experience loss of status for the first time in the U.S., leading to increased financial stress, shifts in gender roles, and greater reliance on partners, all of which can compound their vulnerability to IPV [57,119,130–132].

Our findings also revealed a notable gap in IPV literature examining how structural racism across the phases of migration (e.g., pre-migration, travel, interception) shapes IPV vulnerability among immigrants of color in the US. It is essential for future work to explicitly consider the impacts of exposures to structural racism during various phases of migration, such as persecution, discrimination, and systemic inequities in their home and intermediate countries, may shape immigrants of color's vulnerability to IPV upon arrival in the US [40]. Colonialism has historically disrupted social structures, economies, and cultures in many immigrants' countries of origin [133]. This legacy often results in deep-seated trauma, internalized oppression, and economic marginalization, which can exacerbate power imbalances and stressors in intimate relationships, potentially leading to higher risks of IPV when they resettle in the U.S. [133]. For example, exposure to pre-migration political violence, often linked to the legacy of colonialism, has also been shown to increase the likelihood of US immigrant men's IPV perpetration, thus further underscoring the need for examining structural racism prior to arrival in the US [134]. Additionally, this colonial legacy has contributed to the distrust of authority and services and has created barriers to help-seeking, perpetuating the cycle of violence [133]. These pre-existing exposures to structural racism can influence how individuals navigate and respond to the structural racism they encounter in the US, particularly with law enforcement, and potentially compound their vulnerability to IPV [40]. Such stressors may intersect and further exacerbate vulnerability to IPV through various forms of US-based structural racism, including discriminatory immigration policies, limited access to social services, employment discrimination, and inadequate legal protections [59]. This is also of great concern for women whose immigration status relies on their spouses, where the threat of deportation or loss of legal status may become a tool of control or coercion [59,135]. Future work is needed to safely and ethically collect detailed data on immigration status beyond capturing US-born versus foreign-born distinctions [128,136], as this oversimplification can obscure critical differences in vulnerability to IPV and its harmful repercussions.

Importantly, our findings noted that few studies explicitly discussed community engagement in their work. We identified 5 studies that used community-based participatory research or participatory action research study designs that could provide greater insight into the topic while enabling research participants to contribute to the design of research that addresses needs they observe in their own communities [53,58,65,68,75]. As we continue to advance the IPV field and our understanding of how structural racism impacts IPV among immigrants of color, it is essential that IPV researchers work in partnership with communities [137]. Immigrants of color face multiple barriers to participation in research, whether it is participating in research activities or guiding and developing research efforts [137]. Such barriers not only lead to the underrepresentation of immigrant communities of color in IPV research, but also perpetuate the absence of culturally tailored interventions, policies, and practices; thus, inadvertently may be contributing to the persistently high levels of IPV and harmful repercussions in US-based immigrant communities of color. Our findings highlight significant gaps in the types of immigrant statuses and countries of origin represented in current IPV literature. Future IPV research should prioritize participatory research efforts that engage immigrants of color as active collaborators [137,138]. In concert with these efforts, researchers must embrace reflexivity as a critical practice to advance equity-driven IPV research [139]. Reflexivity requires researchers to continuously reflect on their own positionality, biases, and the power dynamics inherent in their research and relationships with immigrant communities of color [139]. By acknowledging the ways in which their own perspectives shape the research process, IPV and immigration researchers can reduce unintentional harm and work toward more equitable and culturally responsive research outcomes [139]. However, IPV researchers must engage in critical reflexivity as an action-oriented process that informs and transforms their research practices and not simply as an exercise in self-reflection [139]. Research that is led by IPV scholars of color is also of critical importance. These approaches not only enhance the integrity of the research but also fosters more genuine and mutually beneficial partnerships with immigrant communities of color [136–138].

The findings from this scoping review should be interpreted in light of its limitations. As there is no universally agreed-upon definition of structural racism, the conceptualization of structural racism used in this scoping review may reflect the researchers' perspectives and may differ from other definitions. The variation in how structural racism is

defined, operationalized, and attributed across studies limited our ability to identify and discuss patterns in how structural racism drives IPV experiences and consequences among immigrants of color. Therefore, our review focused on whether and how studies named, operationalized and attributed structural racism as a determinant of IPV experiences and consequences rather than exploring broader patterns. In the future, as research in this area advances, we may be able to provide a more comprehensive synthesis of the relationship between structural racism and IPV. Additionally, this review only included literature that was published in peer-reviewed journals, excluding gray literature on structural racism and IPV from community-based, government, or nonprofit organizations. Potentially relevant articles may have been excluded because they did not include search terms in the title or abstract. Importantly, while this review focused on male-perpetrated IPV against women and girls due to its disproportionate impact on immigrant women, other forms of IPV including same-sex IPV and IPV perpetrated against other gender identities could also be driven in part by structural racism. Future work is needed that center immigrant communities of color who also represent diverse gender and sexual orientations. Lastly, there may be potential publication bias, where studies that explicitly name or address structural racism might be less likely to be published [117,118], potentially skewing the review's findings on articles explicitly naming structural racism in IPV literature. Publication bias that results in studies that name structural racism facing higher barriers to publication could play an unmeasured role in explaining the gap in IPV literature examining how structural racism shapes IPV vulnerability.

## Conclusion

This scoping review serves as a call to action for IPV researchers to integrate structural racism as a fundamental component in the analysis of IPV among immigrant communities of color and offers a roadmap for future work. Such efforts are well-aligned with the US National Action Plan to End GBV and could guide future iterations of the plan by outlining the role of structural racism in IPV and the need to incorporate it as a driver in future research, policies, and programs. To advance our understanding of IPV among US-based immigrant communities of color, it is essential to explicitly define, measure, examine, attribute, and name the influence of structural racism, within intersectional and decolonial frameworks that recognize how power operates relationally across systems. Naming structural racism is not solely a semantic choice; it is an act of epistemic justice toward ensuring the systems of power driving inequities are rendered visible and, therefore, open to accountability. When structural racism is left unnamed, it remains unmeasured, underexamined, and unchallenged, allowing its effects to persist unchecked across research, policy, and practice. Without doing so, important elements of IPV experiences and consequences among immigrants of color may be missed within intervention, policy, and practice, and high levels of IPV and harmful repercussions will persist.

## Supporting information

**S1 PRISIMA Checklist. PRISIMA-ScR Checklist.** Reproduced from: Tricco AC, Lillie E, Zarin W, O'Brien KK, Colquhoun H, Levac D, et al. PRISMA Extension for Scoping Reviews (PRISMA-ScR): Checklist and Explanation. Ann Intern Med. 2018;169:467–473. https://doi.org/10.7326/M18-0850. Licensed under the Creative Commons Attribution 4.0 International License (CC BY 4.0).
(DOCX)

**S1 Table. Key findings of the articles included in the scoping review.**
(DOCX)

## Author contributions

**Conceptualization:** Samantha Kanselaar, Sameera S. Nayak, Rochelle R. Davidson Mhonde, Jhumka Gupta.

**Data curation:** Samantha Kanselaar, Areej Khan, Kyle Machicado.

**Formal analysis:** Samantha Kanselaar, Sameera S. Nayak, Rochelle R. Davidson Mhonde, Kyle Machicado, Jhumka Gupta.

**Funding acquisition:** Samantha Kanselaar, Jhumka Gupta.

**Methodology:** Samantha Kanselaar, Sameera S. Nayak, Rochelle R. Davidson Mhonde, Jhumka Gupta.

**Supervision:** Sameera S. Nayak, Rochelle R. Davidson Mhonde, Jhumka Gupta.

**Visualization:** Samantha Kanselaar, Jhumka Gupta.

**Writing – original draft:** Samantha Kanselaar, Sameera S. Nayak, Rochelle R. Davidson Mhonde, Jhumka Gupta.

**Writing – review & editing:** Samantha Kanselaar, Sameera S. Nayak, Rochelle R. Davidson Mhonde, Areej Khan, Kyle Machicado, Jhumka Gupta.

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
