## [Decision Letter · Decision Letter 0]

9 Oct 2025

PGPH-D-25-01556

Insufficient: A scoping review of structural racism and intimate partner violence in US-based immigrant communities of color

Dear Dr. Kanselaar,

Thank you for submitting your manuscript to PLOS Global Public Health. After careful consideration, we feel that it has merit but does not fully meet PLOS Global Public Health’s publication criteria as it currently stands. Therefore, we invite you to submit a revised version of the manuscript that addresses the points raised during the review process.

We look forward to receiving your revised manuscript.

Kind regards,

Zahra Zeinali, MD MPH DrPH

Academic Editor

Journal Requirements:

i. State the initials, alongside each funding source, of each author to receive each grant. For example: "This work was supported by the National Institutes of Health (####### to AM; ###### to CJ) and the National Science Foundation (###### to AM)."

ii. State what role the funders took in the study. If the funders had no role in your study, please state: “The funders had no role in study design, data collection and analysis, decision to publish, or preparation of the manuscript.”

2. Please provide separate figure files in .tif or .eps format.

Additional Editor Comments (if provided):

Thank you for submitting your manuscript to PLoS GPH.

Please review the comments below and the comments from the two reviewers and consider them for your revision.

This is an important and timely manuscript addressing the intersection of structural racism, gender-based violence, and immigrant health in the United States. However, the paper would benefit from deeper theoretical engagement with structural and intersectional frameworks, clearer articulation of inclusion/exclusion logic when expanding the two-tier system, and stronger implications for both research and practice. Below, I offer detailed comments.

1. Theoretical framing

While the manuscript introduces structural racism and intersectionality, it remains somewhat descriptive rather than analytical. There is an opportunity to deepen engagement with critical race and feminist theory:

Integrate conceptualizations of structural racism as a relational system of power (Bailey et al., 2021; Pirtle, 2020; Krieger, 2023).

Situate the analysis within feminist and decolonial frameworks (Crenshaw, Collins & Bilge, Davis, Mohanty) to emphasize how the failure to name structural racism perpetuates epistemic injustice and marginalization of immigrant women’s experiences.

2. Methodological transparency

Clarify the decision-making process behind expanding inclusion into two tiers. How was the threshold defined between explicit and implicit references to structural racism? Was interrater reliability assessed, and how were discrepancies resolved?

The extraction and synthesis appear to have been performed by a single reviewer. Including a brief description of quality assurance measures (e.g., verification, calibration) would strengthen credibility.

3. Analysis and interpretation

The Tier 1/Tier 2 binary, while useful, risks oversimplifying epistemic complexity. Some Tier 2 studies may reflect structural silencing rather than conceptual absence. A brief reflection on this would enhance the critical depth of the discussion.

The results section could benefit from synthesizing differences across immigrant groups (e.g., Latinx, Asian, African, Arab) and by form of IPV or structural factor, even narratively.

4. Implications and future directions

The discussion should go beyond a “call to action” by offering concrete pathways:

How can IPV research explicitly operationalize structural racism (e.g., indicators related to policy, enforcement, segregation, or economic exclusion)?

How can national data systems and funders integrate anti-racist measures?

What are the implications for practice and policy (e.g., survivor services, prevention frameworks)?

Connecting findings to the US National Action Plan to End GBV could anchor the paper’s policy relevance.

5. Literature scope

Include recent works on measuring and conceptualizing structural racism in public health (e.g., Hardeman et al., 2022; Krieger et al., 2023; Gee & Ford, 2011).

Expand feminist and intersectional citations to foreground the intellectual lineage of this critique.

Reviewers' comments:

Reviewer's Responses to Questions

**Comments to the Author**

1. Does this manuscript meet PLOS Global Public Health’s publication criteria ? Is the manuscript technically sound, and do the data support the conclusions? The manuscript must describe methodologically and ethically rigorous research with conclusions that are appropriately drawn based on the data presented.? Is the manuscript technically sound, and do the data support the conclusions? The manuscript must describe methodologically and ethically rigorous research with conclusions that are appropriately drawn based on the data presented.

Reviewer #1: No

Reviewer #2: Yes

2. Has the statistical analysis been performed appropriately and rigorously?

Reviewer #1: N/A

Reviewer #2: N/A

3. Have the authors made all data underlying the findings in their manuscript fully available (please refer to the Data Availability Statement at the start of the manuscript PDF file)?

The PLOS Data policy requires authors to make all data underlying the findings described in their manuscript fully available without restriction, with rare exception. The data should be provided as part of the manuscript or its supporting information, or deposited to a public repository. For example, in addition to summary statistics, the data points behind means, medians and variance measures should be available. If there are restrictions on publicly sharing data—e.g. participant privacy or use of data from a third party—those must be specified.requires authors to make all data underlying the findings described in their manuscript fully available without restriction, with rare exception. The data should be provided as part of the manuscript or its supporting information, or deposited to a public repository. For example, in addition to summary statistics, the data points behind means, medians and variance measures should be available. If there are restrictions on publicly sharing data—e.g. participant privacy or use of data from a third party—those must be specified.

Reviewer #1: Yes

Reviewer #2: Yes

4. Is the manuscript presented in an intelligible fashion and written in standard English?

Reviewer #1: Yes

Reviewer #2: Yes

5. Review Comments to the Author

Reviewer #1: For this article, a scoping review of the literature was conducted to understand the impact of structural racism and IPV on immigrants of color. While the purpose of the review is important and would fulfill a gap in the literature, I find that the scoping review is underdeveloped. There are inconsistencies between the aims of the review and the review strategy for the article. Furthermore, there is confusion on whether the authors really only examined structural racism, as some of the terms are capturing other forms of racism, discrimination, and oppression. Finally, the results section seems severely underdeveloped, as the authors merely provided a broad and brief summary of the articles, instead of any examination or analysis. Though it’s a scoping review, a discussion of how the articles relate to the aims of the review, including definitions, consequences, gaps, etc., would strengthen this manuscript. On a smaller note, there are inconsistencies with spacing and formatting as well.

Introduction

In the introduction, the authors say that the immigrant health advantage erode due to systemic inequalities and cumulative impact of inequities. However, it would be helpful to discuss the systematic inequalities and inequities that the authors are referring to in this sentence and to describe how they lead to worsen health for immigrants.

The second paragraph of the introduction seems undeveloped. There are a bunch of ideas about systemic oppression, the clumping of diverse groups into one group called people of color, and how these all lead to health disparities. However, all together, the topics are glossed over and would be better understood by explaining these relationships further.

The paragraph beginning with, “The interplay of these intersecting identities and…” feels repetitive. It could be woven into the paragraphs before in the introduction.

Methods

Throughout the manuscript, there seems to be differences in spacing within and between paragraphs. Please make spacing consistent throughout.

How were discrepancies resolved by the team?

After a full text review using the search criteria, the authors state that only one person has met the criteria. Therefore, the authors widened the search criteria. When the authors widened the search criteria, did the authors return back to the abstract review? If not, did the authors start the full text search again? Was it blinded again? What was this process like of changing the criteria in the middle of the review?

In the introduction, the authors stated that the authors wanted to discuss IPV as being rooted in systemic inequalities, instead of solely being a product of systemic inequalities. However, it seems as though IPV was named as a consequence to structural factors in Tier 2. Hence, it seems that allowing articles in Tier 2 goes against one of the aims of the scoping review. Can the authors rectify this?

Similarly to my previous comment, Tier 2 also seems less focused on structural racism and, rather, discuss structural barriers, structural inequalities, or structural barriers. I am struggling with the categorizing of all of the terms for structural racism, as there may be other systems of oppression, power, and discrimination at play other than racism. For example, a lot of what the authors discuss can actually be categorized as “Systemic or institutional xenophobia”.

Table 3 is unreadable and cut off for me. It would be better if it was turned so that we can see the entire table. However, in this table, it would be helpful to specify the information that the articles did provide for the important attribute. Otherwise, it is difficult to understand how the authors assessed fit if, for example, key attributes weren’t specifically discussed.

It would be helpful for the authors to explain “narrative synthesis.” Also, it would be helpful for the authors to describe why it was suitable for this scoping review.

Results

I am a little confused why an article was not included because it discussed another type of racism, even though the authors’ definition and operationalization of structural racism is broad and have included terms like ‘police brutality’, which could easily be categorized as racial profiling.

I am a bit confused about how fear of being deported or of deportation is a theme of structural racism.

For tier 1: The authors state that the article discusses intersectionality with an acknowledgement to structural racism, which tells me that structural racism was not the main focus of the article. Again, this is confusing as the purpose of the scoping review lead me to believe that structural racism should have been one of the main foci of the articles included in the review.

The results section is severely lacking. It does not integrate the articles and merely summarizes across the articles very broadly. A deeper discussion of gaps, findings, key definitions, differences by immigrant communities, and other key findings that would extend the literature. The authors did not critically examine the articles. The scoping review seems incomplete.

Discussion

The first sentence of the discussion says this is “one of the first reviews to examine….”. However, I do not believe the articles were critically examined.

The discussion is the most thorough part of the manuscript, with rich information about recommendations.

Reviewer #2: ntroduction

• The definition of structural racism (lines 76–83) could be sharpened to better distinguish it from related terms such as systemic and institutional racism, given the manuscript’s emphasis on conceptual clarity.

• The focus on male-perpetrated IPV against women is justified but somewhat restrictive; the authors should explicitly acknowledge exclusion of other gender identities and same-sex IPV to avoid reinforcing a heteronormative framing.

Methods

• The criteria distinguishing Tier 1 and Tier 2 could be explained more precisely. For example, were articles that used “institutional racism” or “systemic racism” but not “structural racism” considered Tier 1 or Tier 2?

• The Methods section would benefit from a decision tree or supplemental table showing how terms like “legal violence” or “structural violence” were coded relative to structural racism.

• Tables 2 and 3 are very dense; moving detailed study characteristics to the appendix and keeping a summary in-text would improve readability.

Results

• Some numbers in text vs. PRISMA figure may need reconciliation for consistency. Please review the numbers

• The disproportionate focus on Latina/o groups should be emphasized more strongly as a structural issue in the literature itself (i.e., systemic underrepresentation of African, Arab, and Asian immigrant populations).

Discussion

• Recommendations could be made more actionable. For example, suggest specific measures (e.g., redlining indices, immigration enforcement intensity, housing discrimination data) that could be linked to IPV outcomes.

• The discussion of community-engaged research would be stronger if linked back to specific included studies that exemplified this approach.

• Limitations section is appropriate, but the authors should expand on the potential publication bias (studies that name racism may face higher barriers to publication).

6. PLOS authors have the option to publish the peer review history of their article (what does this mean? ). If published, this will include your full peer review and any attached files.). If published, this will include your full peer review and any attached files.

**Do you want your identity to be public for this peer review?** For information about this choice, including consent withdrawal, please see our Privacy Policy ..

Reviewer #1: No

Reviewer #2: No

Figure Resubmissions:

After uploading your figures to PLOS’s NAAS tool - https://ngplosjournals.pagemajik.ai/artanalysis, NAAS will process the files provided and display the results in the "Uploaded Files" section of the page as the processing is complete. If the uploaded figures meet our requirements (or NAAS is able to fix the files to meet our requirements), the figure will be marked as "fixed" above. If NAAS is unable to fix the files, a red "failed" label will appear above. When NAAS has confirmed that the figure files meet our requirements, please download the file via the download option, and include these NAAS processed figure files when submitting your revised manuscript.>

---

## [Decision Letter · Decision Letter 1]

17 Feb 2026

PGPH-D-25-01556R1

Insufficient: A scoping review of structural racism and intimate partner violence in US-based immigrant communities of color

Dear Dr. Kanselaar,

Thank you for submitting your manuscript to PLOS Global Public Health. After careful consideration, we feel that it has merit but does not fully meet PLOS Global Public Health’s publication criteria as it currently stands. Therefore, we invite you to submit a revised version of the manuscript that addresses the points raised during the review process.

We look forward to receiving your revised manuscript.

Kind regards,

Zahra Zeinali, MD DrPH

Academic Editor

Journal Requirements:

Additional Editor Comments (if provided):

Dear Dr. Kanselaar and colleagues,

Thank you for submitting the revised version of your manuscript. The revision has significantly improved the paper’s conceptual clarity, methodological transparency, and interpretive depth. Notably, the revised Discussion section more clearly frames “naming structural racism” as an epistemic justice issue and explains how implicit treatment of structural conditions, without explicit attribution, constrains measurement, accountability, and intervention design.

The manuscript’s central finding remains clear and significant: of 1,896 retrieved records, 32 studies met inclusion criteria in the tiered framework, and only one study explicitly named and attributed structural racism (systemic racism) as a driver of IPV-related experiences or consequences among US-based immigrant communities of color. This finding supports your conclusion that substantial gaps persist in defining, operationalizing, attributing, and naming structural racism in this literature, and that advancing beyond solely individual or cultural framings is essential for the field.

We also acknowledge the improved limitations section, which now addresses (i) definitional contestation, (ii) exclusion of gray literature, (iii) potential publication bias affecting studies that explicitly name structural racism, and (iv) the importance of including diverse gender and sexual orientations in future research.

Decision: Minor Revision

Your manuscript is nearing acceptance. Before proceeding, please address the following final points:

1. Please add a concise summary of the tiering decision rules (in an in-text box or supplement) clarifying the criteria for (a) explicit naming, (b) attribution, and (c) implicit alignment terms (such as legal violence or structural violence). This addition will enhance reproducibility for borderline classifications.

2. To further strengthen the rigor of your study without contradicting your epistemic stance, please include a specific description of the initial classification discordance and its resolution. As you note, eight articles were initially categorized differently; a single sentence in the Methods or Limitations section will suffice.

3. Please copyedit the Discussion and Conclusion sections to address minor textual artifacts, such as spacing issues and typographical errors (for example, “epistemic justicea” and “towarda…ensuringe”).

4. During proofing, please confirm consistency between the PRISMA figure and the text, ensuring that counts and exclusions match exactly throughout.

5. For improved readability, it is recommended (though optional) to consider splitting or relocating the densest table content (Tables 2 and 3) to supplementary material, while retaining a concise summary table in the main text.

Please submit a clean revised manuscript along with a brief response-to-editor letter addressing each of the points listed above.

Reviewers' comments:

Reviewer's Responses to Questions

**Comments to the Author**

1. If the authors have adequately addressed your comments raised in a previous round of review and you feel that this manuscript is now acceptable for publication, you may indicate that here to bypass the “Comments to the Author” section, enter your conflict of interest statement in the “Confidential to Editor” section, and submit your "Accept" recommendation.

Reviewer #2: All comments have been addressed

2. Does this manuscript meet PLOS Global Public Health’s publication criteria ? Is the manuscript technically sound, and do the data support the conclusions? The manuscript must describe methodologically and ethically rigorous research with conclusions that are appropriately drawn based on the data presented.? Is the manuscript technically sound, and do the data support the conclusions? The manuscript must describe methodologically and ethically rigorous research with conclusions that are appropriately drawn based on the data presented.

Reviewer #2: Yes

3. Has the statistical analysis been performed appropriately and rigorously?

Reviewer #2: N/A

4. Have the authors made all data underlying the findings in their manuscript fully available (please refer to the Data Availability Statement at the start of the manuscript PDF file)?

The PLOS Data policy requires authors to make all data underlying the findings described in their manuscript fully available without restriction, with rare exception. The data should be provided as part of the manuscript or its supporting information, or deposited to a public repository. For example, in addition to summary statistics, the data points behind means, medians and variance measures should be available. If there are restrictions on publicly sharing data—e.g. participant privacy or use of data from a third party—those must be specified.requires authors to make all data underlying the findings described in their manuscript fully available without restriction, with rare exception. The data should be provided as part of the manuscript or its supporting information, or deposited to a public repository. For example, in addition to summary statistics, the data points behind means, medians and variance measures should be available. If there are restrictions on publicly sharing data—e.g. participant privacy or use of data from a third party—those must be specified.

Reviewer #2: Yes

5. Is the manuscript presented in an intelligible fashion and written in standard English?

Reviewer #2: Yes

6. Review Comments to the Author

Reviewer #2: NA

7. PLOS authors have the option to publish the peer review history of their article (what does this mean? ). If published, this will include your full peer review and any attached files.). If published, this will include your full peer review and any attached files.

**Do you want your identity to be public for this peer review?** For information about this choice, including consent withdrawal, please see our Privacy Policy ..

Reviewer #2: No

 Figure Resubmissions:

---

## [Editor Report · Decision Letter 2]

19 Mar 2026

Insufficient: A scoping review of structural racism and intimate partner violence in US-based immigrant communities of color

PGPH-D-25-01556R2

Dr. Kanselaar,

We are pleased to inform you that your manuscript 'Insufficient: A scoping review of structural racism and intimate partner violence in US-based immigrant communities of color' has been provisionally accepted for publication in PLOS Global Public Health.

Best regards,

Zahra Zeinali, MD MPH DrPH

Academic Editor

Dear Dr. Kanselaar and colleagues,

Thank you for submitting the revised manuscript (R2), “Insufficient: A scoping review of structural racism and intimate partner violence in US-based immigrant communities of color.” We appreciate the care taken in responding to the editor and reviewer feedback and the clarity of your point-by-point response.

After review of the revised submission and your response, I am pleased to inform you that the manuscript is accepted, pending minor production-level checks.

Minor production checks before finalization:

1. Table readability/layout: Please work with production to ensure Table 3 is fully legible in the final typeset PDF (e.g., landscape orientation, splitting across pages, or moving the detailed version to Supporting Information).

2. Final proof consistency: During proofs, please re-check that PRISMA counts match the text and that spacing remains consistent throughout.

Thank you again for choosing PLOS Global Public Health. We look forward to publishing your work.